# Multi-Scale Fusion Localization Based on Magnetic Trajectory Sequence

**DOI:** 10.3390/s23010449

**Published:** 2023-01-01

**Authors:** Zhan Jin, Ruiqing Kang, Hailu Su

**Affiliations:** School of Automation and Electrical Engineering, University of Science and Technology Beijing, Beijing 100083, China

**Keywords:** indoor positioning, magnetic fingerprint, trajectory extraction, multi-scale features

## Abstract

Magnetic fingerprint has a multitude of advantages in the application of indoor positioning, but as a weak magnetic field, the dynamic range of the data is limited, which exerts direct influence on the positioning accuracy. Aiming at resolving the problem wherein the indoor magnetic positioning results tremendously rest with the magnetic characteristics, this paper puts forward a method based on deep learning to fuse the temporal and spatial characteristics of magnetic fingerprints, to fully explore the magnetic characteristics and to obtain stable and trustworthy positioning results. First and foremost, the trajectory of the acquisition area is extracted by adopting the ameliorated random waypoint model, and the simulation of pedestrian trajectory is completed. Then, the magnetic sequence is obtained by mapping the magnetic data. Aside from that, considering the scale characteristics of the sequence, a scale transformation unit is designed to obtain multi-scale features. At length, the neural network self-attention mechanism is adopted to fuse multiple features and output the positioning results. By probing into the positioning results of dissimilar indoor scenes, this method can adapt to diverse scenes. The average positioning error in a corridor, open area and complex area reaches 0.65 m, 0.93 m and 1.38 m respectively. The addition of multi-scale features has certain reference value for ameliorating the positioning performance.

## 1. Introduction

With the advent of intelligent information life, accurate and trustworthy indoor positioning has a wide range of application scenarios [1]. The mature and perfect satellite positioning system can provide all-weather, all-time, high-precision outdoor positioning for global users, and play a paramount role in regional information searches, traffic management, vehicle navigation and other scenarios [2]. Nonetheless, in the complex indoor environment, as a result of factors such as signal shielding and occlusion, the wireless signal attenuation of satellite positioning is critical and cannot form an effective satellite positioning result. As a result, it is imperative to study effective indoor positioning methods.

Traditional indoor positioning methods can be categorized into active and passive positioning methods in line with whether they use external sources. Common active positioning methods include ultra-wide band (UWB) [3], radio frequency identification (RFID) [4], Wi-Fi [5,6], Bluetooth [7] and methods based on computer vision and visible light [8,9]. Passive location method predominantly rest with inertial measurement unit and magnetic fingerprinting [10]. Considering the above positioning methods, active positioning needs additional auxiliary equipment to be laid, which heightens the positioning cost and is not advantageous for the expansion of indoor space. The interference of the building structure on the source signal cannot be ignored. The positioning method of inertial measurement is based upon the integration of inertial measurement units. On account of the random error and drift error of the sensor, the cumulative error of long-standing positioning cannot be avoided [11]. Meanwhile, the dependence of this method on the initial position makes it often unable to independently locate. In contrast, the indoor positioning method based on magnetic fingerprint can overcome these limitations well. First and foremost, as a natural magnetic phenomenon, magnetic positioning can be completed without adding additional auxiliary equipment. Aside from that, the results of magnetic positioning are not limited by building structure [12].

A complex building structure makes the difference of magnetic fingerprint more apparent and fingerprint characteristics more conspicuous and more beneficial for positioning. Another paramount point is that magnetic fingerprintingcan be easily integrated with other methods to assist positioning. So the study of magnetic positioning is receiving more and more attention [13,14].

The crucial point to indoor positioning using magnetic fingerprint is discovering more features bound up with location information from magnetic data with limited dynamic range and then outputting stable and trustworthy positioning results. In order to heighten the positioning accuracy, the time-series characteristics of magnetic are considered. The positioning analysis is carried out through a battery of magnetic sequences at continuous positions. By considering the time correlation between continuous inputs, the previous method based on spatial input is improved [15]. The LocateMe positioning system uses the Dynamic Time Warping (DTW) algorithm to match the magnetic sequence, and the positioning accuracy is about 2 m [16]. Although the matching algorithm based upon sequence difference makes up for the error of single point matching by time correlation, it also has some limitations. To start with, the sequence length imposes a direct influence on the positioning accuracy and computational complexity. As a consequence, it is imperative to ameliorate and balance the correlation between them. Apart from that, the step length of the sequence sliding window is a paramount factor affecting the accuracy and computation, and when the database information is large, excessive matching heightens the probability of false matching results [17,18]. The DTW algorithm is improved by the adaptive matching of the window size and sequence initial point selection. Research has shown that this method can elevate the search efficiency and positioning accuracy in areas where magnetic characteristics change strikingly, but the positioning performance is noticeably lessened in areas with fuzzy characteristics [19]. For the sake of obtaining the deep features of magnetic fingerprints, machine learning has gradually been extensively employed by researchers. The spatial division of indoor positioning area is a research direction. The area is divided in accordance with the variation characteristics of disturbances in buildings, and the positioning accuracy is elevated by hierarchical positioning. It can also ameliorate the model to explore deeper features. The neural network model is employed as a weak predictor. Multiple groups of weak predictors are integrated into a strong predictor through the ensemble learning method, and the position prediction is completed through ensemble learning [20]. A study put forth a network model that combines the time and space representations of magnetic signal sequences to infer the location features. A convolution neural network (CNN) and recurrent neural network (RNN) were employed to extract features respectively, and the features are fused to make them more discriminative and to achieve more accurate positioning [21]. Some teams adopted dissimilar neural network architectures, such as A CNN and long short-term memory(LSTM), and, ultimately, integrated their results through independent training to obtain the probability map of the user location [22]. Some scholars also built an LSTM network to carry out real-time indoor positioning and proposed a dimension expansion scheme using double sliding windows to preprocess magnetic data, which had satisfactory positioning performance. Meanwhile, it also gave us great inspiration for the selection and setting of network parameters [23]. By introducing the concept of magnetic landmarks, the region can also be divided, and the indoor landmarks can be matched through the magnetic trajectory [24,25]. Nevertheless, the whole positioning process has high requirements for the accuracy of the original data acquisition. The establishment process of landmarks needs to take the extreme value and mean value into consideration. Further more, the screening conditions are susceptible to errors. A host of previous positioning methods are limited to one-dimensional acquisition path, and the positioning mode lacks the evaluation of two-dimensional space. Meanwhile, the fuzzy characteristics of regional geomagnetism are still the primary factor limiting the positioning accuracy.

In an effort to obtain a positioning method with higher positioning accuracy and better universality, this paper comes up with a method to obtain the magnetic sequence by adopting the random waypoint (RWP) model to simulate the pedestrian trajectory. With respect to the input magnetic sequence, LSTM is employed to extract the magnetic characteristics, and the unit of scale transformation is designed. The original magnetic sequence is categorized into subsequences of dissimilar scales. After the same feature extraction of the subsequence, the feature splicing is carried out. The attention mechanism in the neural network is adopted to materialize the weighted fusion of diverse scale features, and, eventually, the position estimation is completed. This method is advantageous for combining the temporal and spatial characteristics of magnetic through pedestrian trajectory, and multi-scale segmentation also highlights the local characteristics of the indoor magnetic, which makes the magnetic sequence with limited dynamic range more characteristic and more beneficial for positioning.

## 2. Positioning System Structure

The overall positioning process is categorized into two phases, as depicted in Figure 1. Above all, the data acquisition is completed in the training stage, and the data of magnetic and inertial measurement units are obtained by adopting the intelligent sensors in a mobile phone. After the corresponding filtering processing of the original data, the magnetic fingerprint database is formed by mapping with the actual spatial coordinates. The optimized RWP model is adopted to simulate the pedestrian trajectory, and the interpolation model is employed to generate the corresponding magnetic sequence. The magnetic sequence is employed as the model input, and it is categorized into subsequence sets with dissimilar sequence lengths. The scale features are extracted and weighted. Last but not least, the actual position information is employed as the output of the model. In the test phase, users obtain the measured magnetic sequence and input the model for feature extraction, and, ultimately, output the predicted positioning results.

## 3. Magnetic Data Processing and Positioning Model

### 3.1. Magnetic Data Acquisition

For the time being, the common magnetic acquisition methods can be categorized into three kinds: path continuous acquisition, path discrete acquisition and grid single-point acquisition. Dissimilar acquisition methods correspond to diverse, magnetic-based indoor positioning solutions. The grid single-point acquisition method corresponds to a two-dimensional plane space, and the path acquisition method corresponds to one-dimensional corridor. MagPIE and UJIIndoorLoc-Mag [26,27], as public magnetic indoor positioning data sets, provide all kinds of sensor data needed for positioning and their acquisition methods, which can be employed for performance analysis of positioning methods. Aside from using the data provided by the public data set, we also carried out data acquisition in a specific area. Considering the acquisition efficiency, accuracy and richness of information, we adopted the continuous path acquisition based upon the step detection of the collectorand mapped the two-dimensional plane grid with the step position (cf. Figure 2). The sampling frequency depended on the collector’s step frequency. On the basis of this principle, the speedy acquisition of the indoor space magnetic was completed.

The data obtained at the collection point were as follows:(1)$$Data=\left(x,y,M_{x},M_{y},M_{z},M_{a},Acc_{x},Acc_{y},Acc_{z}\right)$$
where (x,y) is the position information, $M_{i},\left(i=x,y,z\right)$ represents the triaxial magnetic data, $M_{a}$ represents the single-point magnetic intensity, and the magnetic intensity is calculated using Equation (2):(2)$$M_{a}=\sqrt{{\left(M_{x}\right)}^{2}+{\left(M_{y}\right)}^{2}+{\left(M_{z}\right)}^{2}}$$

$Acc_{i},\left(i=x,y,z\right)$ represents the triaxial acceleration value. Using the Get Rotation Matrix method provided by Android, the rotation matrix and tilt matrix were calculated by inputting the array of the acceleration sensor and the magnetic sensor, so as to actualize the conversion of magnetic data from the mobile phone coordinate system to the world coordinate system. This method is advantageous for eliminating the numerical difference stemming from the acquisition directions of the equipment at the same sampling point.

### 3.2. Random Waypoint Mobility Model

With regard to the positioning process of the two-dimensional space plane, the acquisition of the time series should not be limited to the one-dimensional acquisition path. For the sake of obtaining a more realistic and universal positioning model, we adopted the random waypoint model to simulate the pedestrian trajectory in the acquisition area so as to obtain the training data and satisfy the needs of deep learning, which usually requires a multitude of training data.

RWP is the fundamental building block for most routing protocols, initially employed to simulate communication protocols such as mobile Ad Hoc networks [28]. In this paper, we used RWP to simulate the trajectory of pedestrian movement in two-dimensional space. The fundamental principle flow chart of RWP is depicted in Figure 3. First and foremost, the size of the location area is presupposed, and a destination coordinate in the area is randomly generated and then moved from the preset starting point to the destination. The moving speed follows the uniform distribution of $V\in \left(Vmax_{\min}\right)$, where $Vmax_{min}$ is the minimum and maximum of the set one-step moving speed. When the moving trajectory reaches the destination, it randomly stays for a period of time, and the pause time follows the uniform distribution $T_{p}\in \left(Tmax_{\min}\right)$. $Tmax_{min}$ is the preset minimum and maximum residence time, and then the next destination coordinate is randomly generated. The original destination is treated as a new starting point, and the above process is repeated until the number of generated trajectory points satisfies the requirements. The moving destination and velocity of each trajectory are independent.

The path coordinate $\left(X_{i},Y_{i}\right)$ generated by RWP is the recorded trajectory point, where $\left(X_{i},Y_{i}\right)$ is obtained as in Equation (3):(3)$$\{\begin{array}{c} X_{i}=X_{i-1}+V\cos(\theta )\\ Y_{i}=Y_{i-1}+V\sin(\theta ) \end{array}$$
where $\left(X_{i-1},Y_{i-1}\right)$ is the trajectory point of the previous step, V is the velocity in the current trajectory, and θ is the angle value of the current motion direction, which is determined by the coordinates of the endpoint and the starting point. The calculation method of θ is expressed as exhibited in Equation (4):(4)$$\theta =\arctan(\frac{Y_{end}-Y_{start}}{X_{end}-X_{start}})$$
where $\left(X_{start},Y_{start}\right)$ and $\left(X_{end},Y_{end}\right)$ are the coordinates of the starting point and the destination point respectively.

The predominant drawback of the RWP model is the non-uniform distribution of the trajectory points, which is spawned from the specific movement behavior. Because the node may choose the central node as the destination end point, or choose the node that can reach the destination through the central area, it will produce the phenomenon of the trajectory converging to the central area. A common improvement measure is the Gaussian Markov model [29], which correlates new speed and new position with the previous speed and position, so as to make up for this deficiency. Nevertheless, the implementation of this model is complex, so it is not extensively employed. The common improvement measures are Gaussian Markov model, by making the new speed and position are correlated with the previous speed and position, to compensate for the uneven distribution. Nonetheless, the implementation of this model is complex, so it is not extensively used. Considering the random direction model similar to RWP, this model randomly selects the direction, then moves to the boundary point of this direction and then selects the next direction. In this way, the center probability density becomes smaller, and the edge probability density becomes larger [30]. Inspired by this, we ameliorated the uneven trajectory distribution by changing the weight at the boundary position in the random generation process of the endpoint.

Figure 4 depicts the trajectory fragment generated by the RWP model, where point A is the preset starting point, and point B, C and D are the subsequent random destination points.

### 3.3. Clough-Tocher Interpolation

In this paper, we used RWP to simulate pedestrian trajectory to generate 100,000 trajectory points. After obtaining the coordinate points of the trajectory, it is essential to map the position information with the actual magnetic data. Nevertheless, single point acquisition based upon the grid cannot ensure that each point on the trajectory has collected the corresponding magnetic data. The scattered distribution of observation data and the unstrict grid distribution affect the accuracy of data processing. As a consequence, it is imperative to select the appropriate numerical interpolation method to encrypt the magnetic database to elevate the numerical accuracy and the final inversion effect.

With respect to the interpolation research of magnetic field, results obtained by the radial basis function method and Kriging method can meet general requirements in accuracy and smoothness. Nonetheless, these methods need to set various parameters, especially the Kriging method. The estimation and calculation of parameters are complex, and unreasonable parameter setting seriously affects the results [31]. As illustrated by the application results of the Clough-Tocher (C-T) method, it can be applied to spatial scattered distribution of potential field data interpolation, and the interpolation results of magnetic data are ideal. Because this method belongs to the local method, it can still maintain high performance in large-scale data, so we used the Clough-Tocher interpolation to fit the magnetic field intensity at any given point.

The open-source algorithm library and mathematical tool SciPy provided the ready-made C-T two-dimensional interpolation model. The magnetic trajectory sequence (GTS) was fitted in line with the trajectory points by modeling the collected magnetic data. The GTS was defined as displayed in Equation (5):(5)$$GTS=\left[\begin{array}{ccccc} X^{1} & Y^{1} & M_{x}^{1} & M_{y}^{1} & M_{z}^{1}\\ \vdots & \vdots & \vdots & \vdots & \vdots \\ X^{t-1} & Y^{t-1} & M_{x}^{t-1} & M_{y}^{t-1} & M_{z}^{t-1}\\ X^{t} & Y^{t} & M_{x}^{t} & M_{y}^{t} & M_{z}^{t} \end{array}\right]\;$$
where *t* represents the length of the magnetic trajectory sequence, (X,Y) represents the trajectory point coordinates, and $M_{x},M_{y},M_{z}$ represent the triaxial magnetic intensity. GTS is the set of magnetic signals recorded by the pedestrian moving track, and its characteristics are more abundant than those of a single magnetic signal.

### 3.4. Long Short-Term Memory Network

RNN has a strong ability to process time series data. Using the advantages of RNN processing time series data to process magnetic fingerprint time series can be more efficient and sufficient for excavateing and extracting the effective positioning features contained in the signal sequence, and ameliorate the positioning performance.

In this paper, the LSTM network model in an RNN was adopted to extract the characteristics of magnetic sequences. In comparison with the traditional RNN model, LSTM can not only memorize the past information, but also selectively forget some unimportant information by adding a gated mechanism to model the long-term temporal relationship.

The elementary structure of the LSTM network is exhibited in Figure 5. The gate structure, including a sigmoid neural network layer and a bitwise multiplication operation, is selective to information. The calculation of the forget gate is as follows:(6)$$f_{t}=\sigma \left(W_{f}\cdot \left[h_{t-1},x_{t}\right]+b_{f}\right)$$$x_{t}$ is the input of the current time, and $h_{t-1}$ is the information stored in the hidden layer of the previous time. After these two vectors are spliced, they are point multiplied with the weight parameter vector $W_{f}$, by activating function, a value between 0 and 1 is obtained. A value of 1 means that the gate is wholly opened; 0 means that the gate is utterly closed, and the other door structure parameters are the same. The input gate is calculated as follows:(7)$$i_{t}=\sigma \left(W_{i}\cdot \left[h_{t-1},x_{t}\right]+b_{i}\right)$$
(8)$${\tilde{C}}_{t}=\tanh(W_{C}\cdot \left[h_{t-1},x_{t}\right]+b_{C})$$

The calculation of the output gate can be expressed as:(9)$$o_{t}=\sigma \left(W_{o}\cdot \left[h_{t-1},x_{t}\right]+b_{o}\right)$$

The calculation of long and short memory is as follows:(10)$$C_{t}=f_{t}*C_{t-1}+i_{t}*{\tilde{C}}_{t}$$
(11)$$h_{t}=o_{t}*\tanh(C_{t})$$

**Figure 5 sensors-23-00449-f005:**
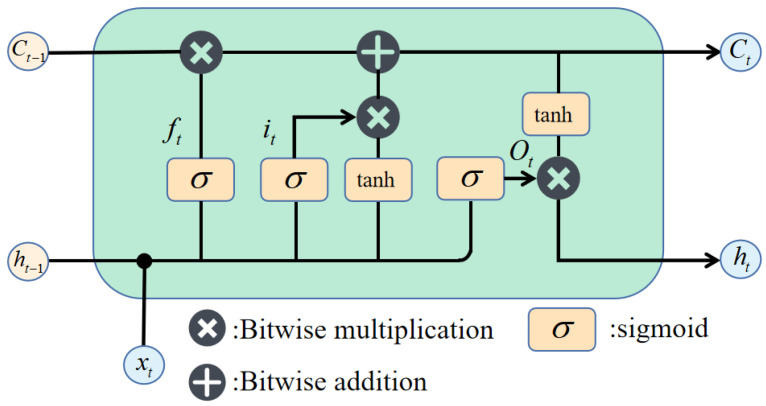
Basic structure of LSTM network.

### 3.5. Multi-Scale Feature Extraction and Fusion

With respect to the input magnetic trajectory sequence, in an effort to extract the characteristics of dissimilar sequence length, it was not only essential to divide the input sequence into diverse length subsequences, but also imperative to take the size of the sequence step length into consideration. Assuming that the input sequence length is N, the reference scale is n, and n is also the step size of the sequence sliding window; the correlation between these two satisfies N = kn, and k is a positive integer. The selection of N is associated with the actual acquisition situation. Since the acquisition of magnetic sequences requires the positioning personnel to carry out a certain distance of mobile acquisition, considering the factors of acquisition distance and sampling frequency and the change of data accuracy triggered by interpolation processing, the sequence length N in this paper was set to 30–50. The selection of n was based upon the local anomaly of the complex indoor environment. Considering that the local anomaly is often accompanied by the characteristics of fast distance attenuation, the length of n corresponds to the length of data collected by the unit foot length of pedestrians.

If M scales are considered, subsequence sets of dissimilar scales are obtained as follows:(12)$$S=\left\{S_{1}\cdots S_{m}\cdots S_{M}\right\}$$
where $S_{m}$ is the subsequence set of sequence length L=m×n, and 1≤m≤M. The acquisition method of the magnetic subsequence set is depicted in Figure 6. In the figure, taking the original sequence length N=200, n=50 and subsequence scale 2n as an example, the subsequence set $S_{2}=\left\{L_{1},L_{2},L_{3}\right\}$ with scale 2n is obtained.

With regard to the segmented subsequence sets of dissimilar scales, LSTM was adopted for feature extraction. In an effort to normalize and linearize, feature splicing, batch normalization, and rectifier linear unit (Relu) were added after the LSTM network, and Dorpout was added after Relu for regularization.

In an effort to find a small amount of crucial information in the magnetic multi-scale sequence and focus on pivotal information, we introduced a self-attention mechanism to capture data features. With regard to the feature vectors extracted from diverse scale magnetic sequences, the distribution of dissimilar feature weight coefficients was materialized by multiplication with the attention value, and then the parameters in the neural network model were determined by self-learning. The overall framework of the positioning process is displayed in Figure 7. Taking the magnetic sequence as the model input, the scale segmentation unit was designed to divide the original magnetic sequence into subsequence sets with dissimilar scales. The LSTM network was employed to extract the features of the subsequences in the subsequence sets of diverse scales, the subsequence features were spliced by Concat method. Then, the attention mechanism was introduced to weight learning for each scale feature. After learning, the overall features of each scale were spliced and fused. Ultimately, in line with the final fusion feature of the neural network, the position coordinates corresponding to the sequence trajectory points were employed as the output of the network to complete the position prediction.

## 4. Experimental Verification and Analysis

In this section, we first introduce the experimental environment and the parameter settings of the positioning model. Subsequently, we explore the positioning results and compare them with some positioning methods in the references.

### 4.1. Experiment Setting

The magnetic characteristics of dissimilar regions are different. For the sake of obtaining a positioning method with high universality, the experiment in this paper will probe into the magnetic positioning results of diverse indoor scenes, including corridors, open areas and complex areas with more office furniture. The public data set MagPIE provided magnetic data from the corridor of the Coordinated Sciences Laboratory (CSL). We first used this data set as the experimental data for corridor positioning, and then selected the spacious area in the office building as the open area. Subsequently, we selected an area with more bookshelves in the library as the complex area for collecting magnetic related data. The size of the three positioning areas was: CSL: 60 m × 15 m; complex area: 20 m × 13 m; open area: 15 m × 7 m. Data acquisition was carried out using the built-in sensor of Android 10 Huawei nova8 mobile Phone, and SQLite database was used to save the acquisition results. Above all, we preprocessed the data set of the CSL laboratory. The original magnetic data acquisition frequency was 50 Hz, which is aligned with the position information in accordance with the time stamp. Considering the large amount of data provided by the data set, the data were sampled to 5 Hz, and the training data and test data were processed in the same way. The processed CSL data set included training trajectory point 26,007 and test trajectory point 7235. The magnetic acquisition points of the complex area and open area were 1040 and 420, respectively. The number of simulated trajectory points was 100,000, and the proportion of training to testing was 3:1. In this experiment, the model parameter settings was as exhibited in Table 1. With the magnetic trajectory sequence as the model input, the initial magnetic sequence length was set to 50, and the reference length of the subsequence was set to 5.

### 4.2. Experimental Results

The multi-scale segmentation of magnetic sequencesdirectly affected the positioning accuracy and computational efficiency of the model. In an effort to take into account the accuracy and complexity of the positioning model, the influence of the scale division of the original sequence on the positioning performance was analyzed first. Taking the corridor data collected by CSL as the analysis object, and taking 100 training times as an example, the training results of several dissimilar scale fusions were compared, and the cumulative distribution function (CDF) of the localization error was as exhibited in Figure 8. As indicated by the blue, solid line in the figure, only the original magnetic sequence was considered as the input, and the other curves added diverse scale subsequences and fused them for positioning. Obviously, after adding the subsequence features of the original sequence, the positioning performance was markedly ameliorated, and the proportion of positioning errors within 1 mincreased from 34.2% to 70%.

The positioning method of the multi-scale magnetic sequence tremendously heightened the positioning accuracy compared with the single original sequence. The error data box plot of the multi-scale fusion is exhibited in Figure 9. The average positioning error before and after considering the multi-scale characteristics was lessened from 2.1 m to about 1.2 m, which immensely ameliorated the positioning performance. Nevertheless, after adding subsequence features, the number of subsequences added slightly heightened the positioning accuracy. Further refining the subsequence set made inconsequential differences in the overall positioning performance, and the model was able to better complete the position estimation. Meanwhile, with the addition of multi-scale subsequences, the complexity of the model was higher and the amount of computation was greater. In an effort to obtain the rich features of the magnetic fingerprint and heighten the operation efficiency of the model, the following experiments took the fusion of three scale features into account so as to complete the position estimation.

With regard to the three indoor scenes of the corridor, open area and complex area, the multi-scale fusion positioning of magnetic sequences was carried out respectively, and the cumulative distribution of the positioning error of the three scenes was as displayed in Figure 10. The average positioning errors of multi-scale fusion positioning in the three regions was 0.65 m, 0.93 m and 1.38 m, respectively, with excellent positioning accuracy. In the public data set, 94% of the samples of the corridor data had a positioning error less than 1 m, and 90.4% had a positioning error of less than 0.6 m. Furthermore, the test results in the open area were close to the data in the corridor, and the error of 91.5% of the test results was controlled within 1 m. Of the positioning results of the test sample of the complex area, 81.8% had an error within 1 m. In comparison with the previous two areas, the positioning performance was slightly decreased, which correlated with the acquisition density of the original data in the complex area. The bookshelf and various office furniture made the data acquisition unable to traverse each point on the positioning plane.

We also explored the loss function values in the training process. Taking the training results of the public data set as an example, the loss function changes of the training set and the verification set data were as displayed in Figure 11. The curves of these two changes were close to each other, and gradually converge with the augmentation of the number of iterations, and eventually converged to about 0.7. The satisfactory convergence process also reflected that the positioning model can effectively learn the characteristics of magnetic sequences.

The heat map of positioning errors was drawn by randomly selecting some test points as depicted in Figure 12. In the randomly selected test points, if the error above 2 m was regarded as a failure of positioning, the probability of correct positioning is 96.3%. As clearly demonstrated by the experimental results, the positioning error on the overall test path remained stable, which also indicates that it is feasible to adopt neural networks for multi-scale fusion positioning in magnetic indoor positioning, and better positioning accuracy can be materialized through limited magnetic characteristics.

### 4.3. Comprehensive Evaluation

In this part, we compare and discuss the method of geomagnetic trajectory sequence multi-scale fusion (GTSMF) employed in this paper with the following common magnetic positioning methods:RNN: References [32,33] used recurrent neural networks to learn location-related features, and used magnetic sequences as input to train a standard RNN network to predict the user’s location. In our experiment, other than the fundamental RNN model, LSTM, GRU and BiLSTM network models were constructed [34]. The RNN model had four layers; LSTM and GRU were two-layer structure; the input sequence length was set to 50, the mini-batch was set to 64; the hidden unit was set to 128; the learning rate was 0.005; and the number of iterations was 100. Taking the data in the public data set MagPIE as the test object, we compared the above method with the proposed method. The comparison of error probability distribution is shown in Figure 13. It can be seen from the graph that the positioning method in this paper is superior to the traditional RNN network model, and the positioning performance of GRU and LSTM is similar. BiLSTM considers future information on the basis of past information and ameliorates positioning performance. The four-layer RNN has poor performance in the training process, slow convergence speed and large positioning error. It also needs to adjust the network parameters and input data. In comparison with the positioning method combined with multiple network architectures, our positioning method obtaind a smaller positioning error and elevated the positioning accuracy by about 10%. Moreover, the direct output of the predicted position information eliminated the subsequent probability data processing and heightened the positioning efficiency [22]. Compared with the method using multi-layer LSTM for real-time positioning, our data acquisition method is not limited to typical trajectory acquisition. In the case of equal positioning accuracy, our positioning method is applicable to dissimilar positioning scenarios, showing better universality [23].

2.DTW: Dynamic time warping matches the measured geomagnetic sequence with the database and finds the matching point with the minimum cumulative distance as the positioning point [35]. When the sequence length and sliding window length of the matching algorithm are the same as the previous ones, the distribution of the positioning error is as exhibited in Figure 13. It was found that the positioning error obtained by DTW is larger than that obtained by machine learning, and the positioning results were also close to the references and our previous experimental results [36]. Owing to the limited dynamic fluctuation range of magnetic data, when the amount of database data was large and the magnetic variation in the positioning area was not conspicuous, the positioning performance was remarkably decreased.3.CNN: Convolutional neural networks have excellent performance in the field of image recognition. A myriad of studies has used CNN to identify geomagnetic maps to infer positions and obtain desirable positioning accuracy. AMID is the first indoor positioning system that used deep neural network to identify magnetic sequence patterns [24], and it achieved 1.7 m positioning accuracy by classifying landmarks. By combining the convolution layer and recursive layer, the positioning accuracy was heightened to 0.95 m [37]. MINLOC [38], materialized the combination of multiple CNNs by voting mechanism, and the average positioning error was about 0.7 m, which is similar to the positioning performance of our method. Reference [38] introduced an attention mechanism to make positioning accuracy superior, achieving a positioning error close to 0.4 m. The desirable positioning performance of CNNs also gave us inspiration. Using reasonable combinations of various network models to explore the deeper characteristics of geomagnetic fingerprints is also our next exploration direction.

## 5. Conclusions

With regard to magnetic indoor positioning, the ambiguity of magnetic signals is the chief reason for the limited the positioning accuracy, which hinders the wide application of positioning. In an effort to explore the deep characteristics of the magnetic sequence, this paper used the RWP model to simulate the pedestrian trajectory, and used the interpolation algorithm to map the trajectory and the magnetic fingerprint to obtain the magnetic trajectory sequence, and then actualized the feature extraction of the magnetic sequence by LSTM. In order to take full account of the local anomalies contained in the magnetic sequence, the scale transformation unit was designed to divide the original sequence into diverse scale subsequences, and the neural network self-attention mechanism was adopted to realize multi-scale feature fusion to obtain more prominent position features and achieve a more accurate position estimation. In order to verify the feasibility of the scheme, this paper conducted experimental verification for three dissimilar indoor regions. As demonstrated by the experimental results, this method can adapt to the positioning of diverse indoor scenes, and multi-scale fusion can effectively heighten the positioning accuracy, which has certain reference significance for exploring sequence characteristics.

## Figures and Tables

**Figure 1 sensors-23-00449-f001:**
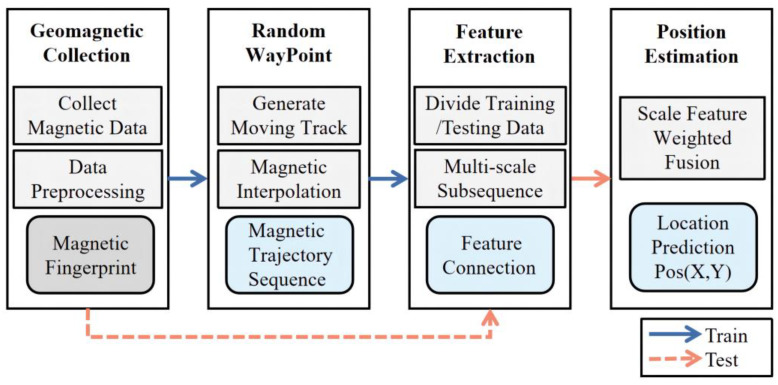
Schematic view of positioning system.

**Figure 2 sensors-23-00449-f002:**
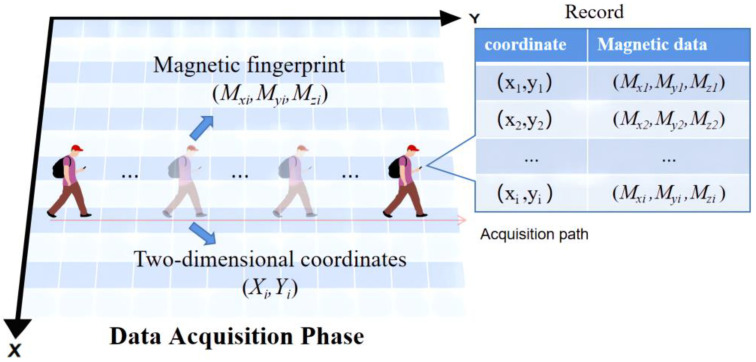
Data acquisition process.

**Figure 3 sensors-23-00449-f003:**
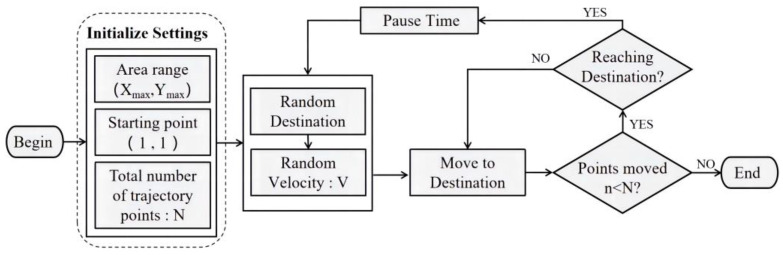
Schematic flow chart of RWP model.

**Figure 4 sensors-23-00449-f004:**
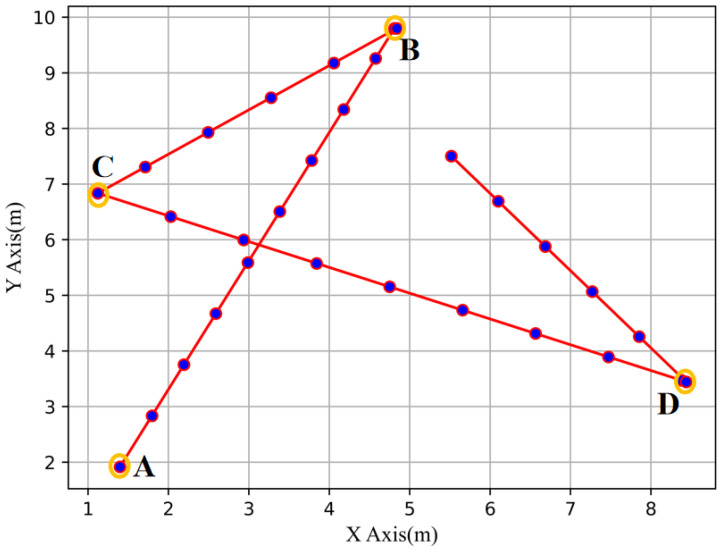
Schematic flow chart of RWP model.

**Figure 6 sensors-23-00449-f006:**
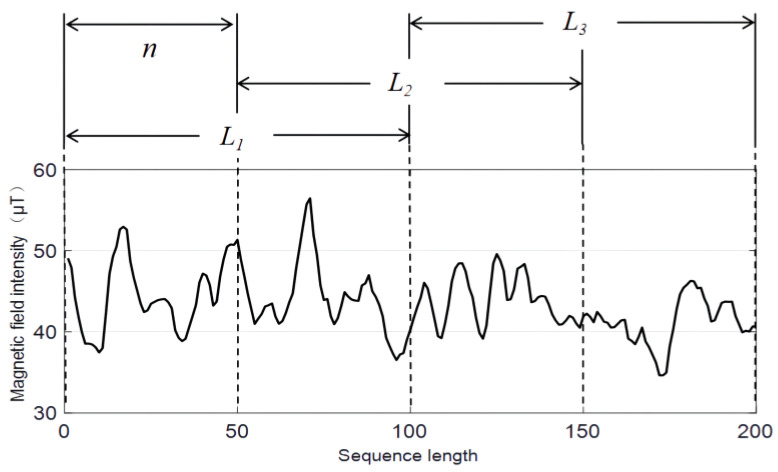
Acquisition of magnetic sequence set.

**Figure 7 sensors-23-00449-f007:**
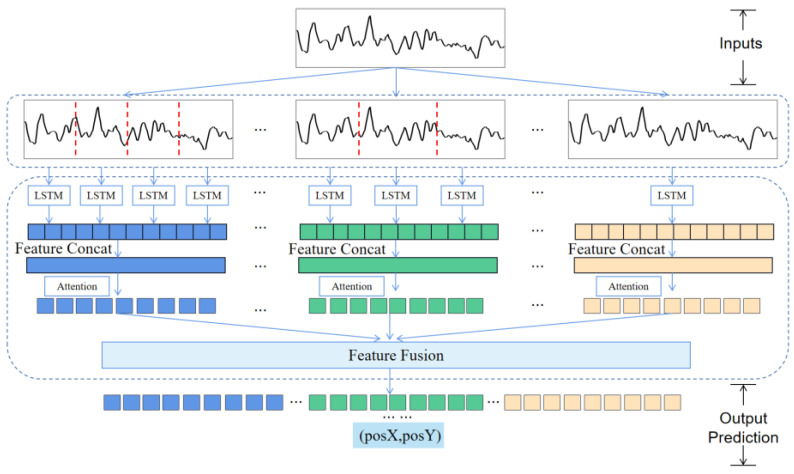
Overall framework of positioning process.

**Figure 8 sensors-23-00449-f008:**
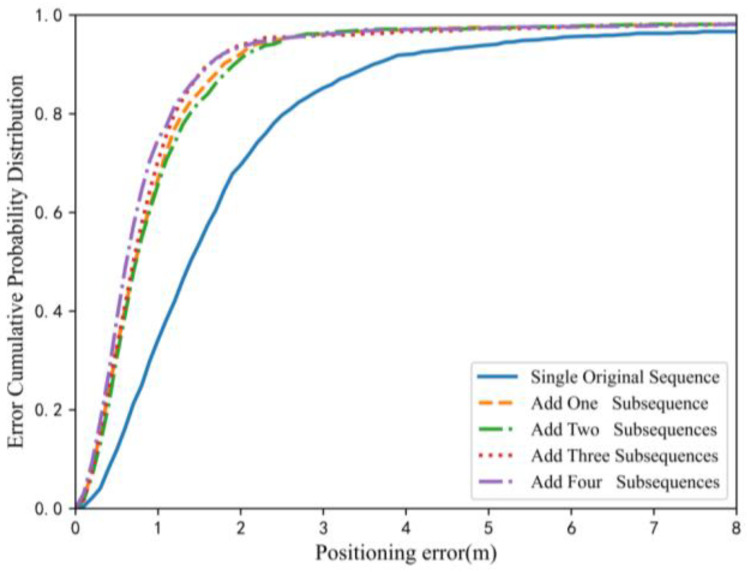
CDF of multi-scale sequence positioning error (the solid line indicates that only the original magnetic sequence was input, and other lines indicate that there were subsequences decomposed from the original sequence apart from the original sequence.).

**Figure 9 sensors-23-00449-f009:**
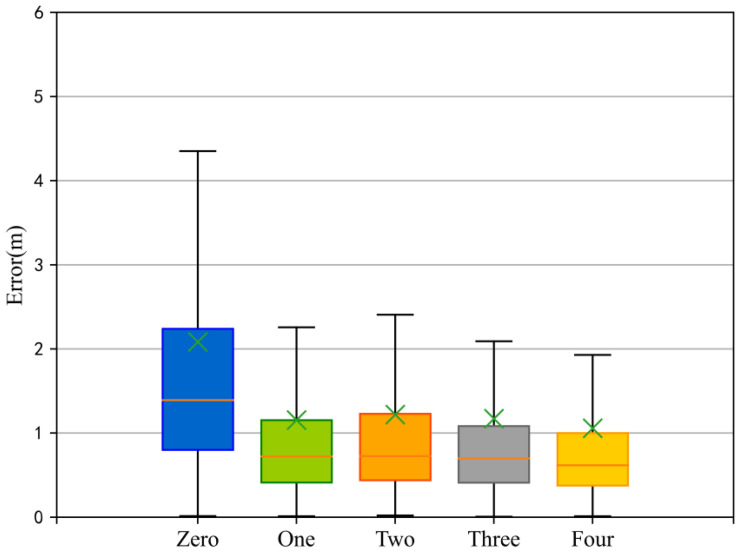
Box plot of multi-scale sequence fusion positioning error (zero represents the positioning result of a single original sequence as input, and other representations fuse the corresponding number of diverse scale subsequences).

**Figure 10 sensors-23-00449-f010:**
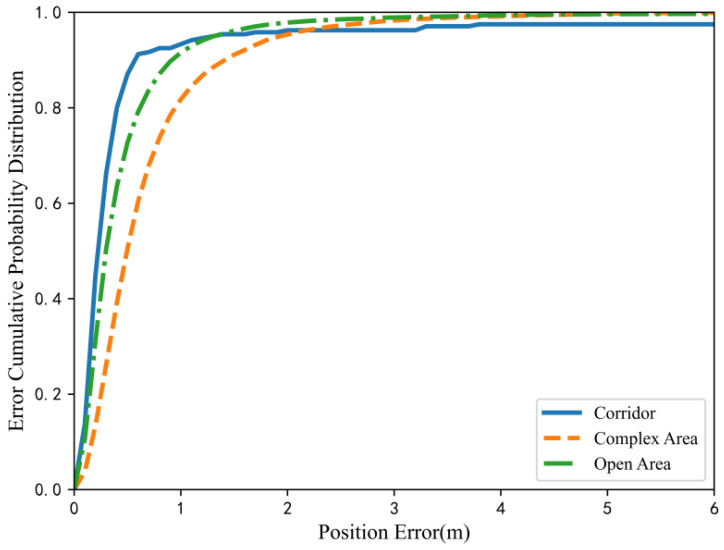
CDF of positioning errors in dissimilar regions.

**Figure 11 sensors-23-00449-f011:**
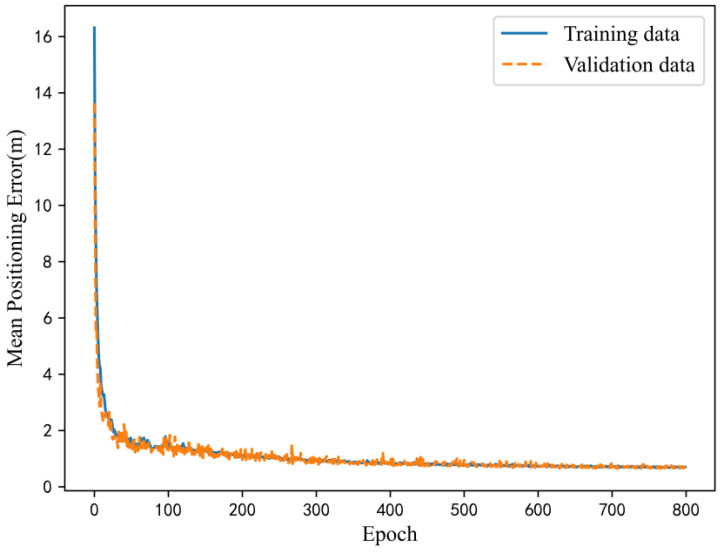
Variation curve of loss function in training process.

**Figure 12 sensors-23-00449-f012:**
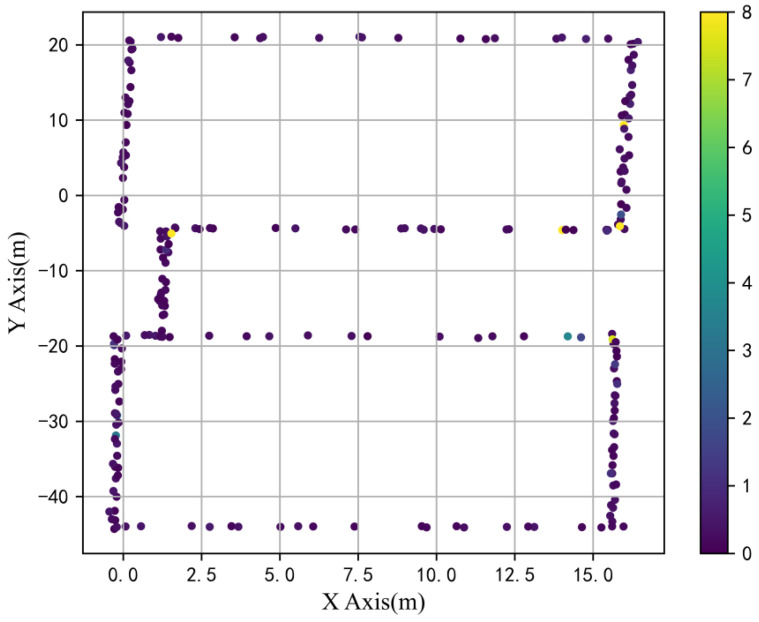
Positioning error heat map of test data on the acquisition path.

**Figure 13 sensors-23-00449-f013:**
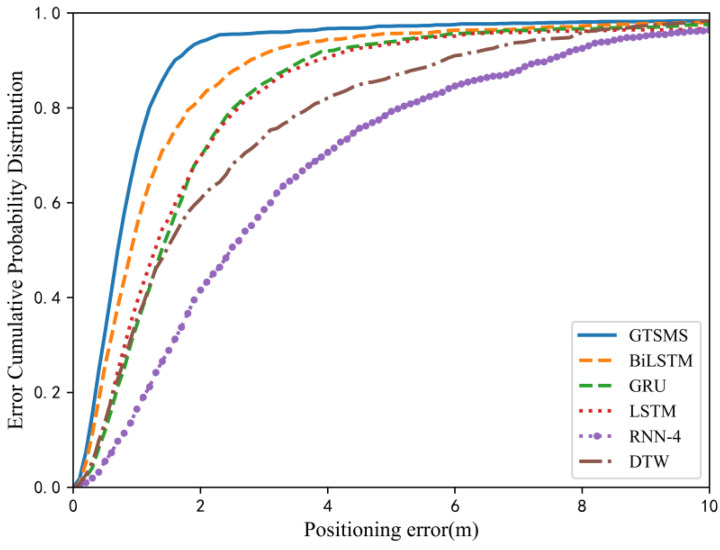
CDF of diverse location methods (GTSMS is the positioning strategy in this paper, and other methods are common positioning models and matching algorithms.).

**Table 1 sensors-23-00449-t001:** Model parameters setting.

Parameter	Parameter Value
loss function	MSE
optimizer learning rate hidden node batch size dropout rate iteration times	Adam 0.005 128 32 0.1 800

## Data Availability

The study did not report any data.

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
