# Peer review of "Multi-Scale Fusion Localization Based on Magnetic Trajectory Sequence"

_sensors, 2023, doi:10.3390/s23010449_

Round 1
Reviewer 1 Report
In this paper, multi-scale fusion localization based on magnetic trajectory sequence was proposed. First, trajectory was extracted from random waypoints, which completes the pedstrain trajectory. Followed by magnetic mapping self-attention neural network was proposed. The authors have provided a solid literature review, and a comprehensive experimental approach was adopted. Thus, based on my review, I recommend this article for publication.
Some minor improvements could be useful:
1) The presentation of this article could be improve, and quality of English must be revised.
2) Quality of Figure 7 could be improved, it's hard to read.
3)Sub-section 4.3 "Discuss" must be replaced with a reasonable word.
Author Response
Dear reviewer and editor,
First of all, thank you very much for your reading and revision of the article, and thank you for your valuable comments. Your suggestions on the content, format, research methods and other aspects of our paper play a very important role in improving the quality of the paper! We have carefully read your comments and carefully made the following modifications item by item:
Point 1: The presentation of this article could be improve, and quality of English must be revised.
Response 1: We have revised the full text in English and improved the expression and English quality through the English editing platform. Thank you very much for your suggestions. Please check whether further optimization is needed. Because the full text has been revised in English, the "Track Changes" function has not been used. In order to facilitate you to view the revised part of the manuscript, I mark the revised part in red. Thank you for your review again. I hope we can complete an excellent paper under your guidance.
Point 2: Quality of Figure 7 could be improved, it's hard to read.
Response 2: We have improved the quality of Figure 7, optimized the details, and improved the interpretation of the picture. Thank you very much for this suggestion.
Point 3: Sub-section 4.3 "Discuss" must be replaced with a reasonable word.
Response 3: Sub-section 4.3 "Discuss" has been replaced by "Comprehensive Evaluation", Please evaluate whether such replacement is appropriate, and thank you for reviewing again.
Reviewer 2 Report
The paper proposes the RWP model to simulate the pedestrian trajectory, and uses the interpolation algorithm to map the trajectory and the magnetic fingerprint to obtain the magnetic trajectory sequence, and then realizes the feature extraction of the magnetic sequence by LSTM. The paper is well written and the approach well described. However, its main drawback is lack of comparative results with other similar methods as list in [1] and [2].
[1] Fernandes, Letícia, et al. "An infrastructure-free magnetic-based indoor positioning system with deep learning." Sensors 20.22 (2020): 6664.
[2] Zhang, Mingyang, et al. "Real-time indoor localization using smartphone magnetic with LSTM networks." Neural Computing and Applications 33.16 (2021): 10093-10110.
Author Response
Dear reviewer and editor,
First of all, thank you very much for your reading and revision of the article, and thank you for your valuable comments. Because the full text has been revised in English, the "Track Changes" function has not been used. In order to facilitate you to view the revised part of the manuscript, I mark the revised part in red. Thank you for your review again. I hope we can complete an excellent paper under your guidance.
Point 1: The paper is well written and the approach well described. However, its main drawback is lack of comparative results with other similar methods as list in [1] and [2].
[1] Fernandes, Letícia, et al. "An infrastructure-free magnetic-based indoor positioning system with deep learning." Sensors 20.22 (2020): 6664.
[2] Zhang, Mingyang, et al. "Real-time indoor localization using smartphone magnetic with LSTM networks." Neural Computing and Applications 33.16 (2021): 10093-10110.
Response 1: Thank you very much for your suggestions. We have carefully read the literature you recommended, and introduced similar methods and made a simple comparison of experimental results in the literature review and Comprehensive Evaluation sections of the article. Thank you for reviewing and revising our revised paper again, and hope that we can complete an excellent paper under your guidance.